# Multiple low dose therapy as an effective strategy to treat EGFR inhibitor-resistant NSCLC tumours

João M. Fernandes Neto [1], Ernest Nadal [2,11], Evert Bosdriesz [1,10,11], Salo N. Ooft [3,11], Lourdes Farre [2,4], Chelsea McLean [3], Sjoerd Klarenbeek[5], Anouk Jurgens[1], Hannes Hagen[1], Liqin Wang[1], Enriqueta Felip[6,7], Alex Martinez-Marti[6,7], August Vidal[2], Emile Voest [3], Lodewyk F. A. Wessels[1], Olaf van Tellingen [8], Alberto Villanueva [2,9] & René Bernards [1✉]

Resistance to targeted cancer drugs is thought to result from selective pressure exerted by a high drug dose. Partial inhibition of multiple components in the same oncogenic signalling pathway may add up to complete pathway inhibition, while decreasing the selective pressure on each component to acquire a resistance mutation. We report here testing of this Multiple Low Dose (MLD) therapy model in *EGFR* mutant NSCLC. We show that as little as 20% of the individual effective drug doses is sufficient to completely block MAPK signalling and pro-liferation when used in 3D (RAF + MEK + ERK) or 4D (EGFR + RAF + MEK + ERK) inhibitor combinations. Importantly, *EGFR* mutant NSCLC cells treated with MLD therapy do not develop resistance. Using several animal models, we find durable responses to MLD therapy without associated toxicity. Our data support the notion that MLD therapy could deliver clinical benefit, even for those having acquired resistance to third generation EGFR inhibitor therapy.

[1] Division of Molecular Carcinogenesis and Oncode Institute. The Netherlands Cancer Institute, Plesmanlaan 121, 1066 CX Amsterdam, The Netherlands. [2] Group of Chemoresistance and Predictive Factors, Subprogram Against Cancer Therapeutic Resistance (ProCURE), ICO, Oncobell Program, IDIBELL, L'Hospitalet del Llobregat, Barcelona, Spain. [3] Division of Molecular Oncology and Immunology, The Netherlands Cancer Institute, 1066 CX Amsterdam, The Netherlands. [4] Institute Gonçalo Moniz, Fundaçao Oswaldo Cruz (FIOCRUZ), Rio de Janeiro, Brasil. [5] Experimental Animal Pathology, The Netherlands Cancer Institute, 1066 CX Amsterdam, The Netherlands. [6] Department of Medical Oncology, Vall d'Hebron University Hospital and Vall d'Hebron Institute of Oncology (VHIO), Barcelona, Spain. [7] Autonomous University of Barcelona (UAB), Barcelona, Spain. [8] Division of Clinical Pharmacology, The Netherlands Cancer Institute, 1066 CX Amsterdam, The Netherlands. [9] Xenopat S.L., Business Bioincubator, Bellvitge Health Science Campus, Barcelona, Spain. [10] Department of Computer Science, Faculty of Science, Vrije Universiteit, Amsterdam, The Netherlands. [11] These authors contributed equally: Ernest Nadal, Evert Bosdriesz, Salo N. Ooft. ✉email: r.bernards@nki.nl

Inhibition of signalling pathways that are activated by onco-genic mutations elicit therapeutic responses due to "addiction" of the cancer to the activated pathway[1]. However, in advanced cancers, development of resistance is practically inevitable due to secondary mutations that restore signalling through the drug-inhibited pathway. Such acquired resistance mutations affect either the drug target itself or components that act upstream, downstream or parallel to the activated signalling component[2,3]. In *BRAF* mutant melanoma and non-small cell lung cancer (NSCLC), inhibition of two components of the same oncogenic pathway (BRAF + MEK, referred to as "vertical targeting") has been shown to provide more lasting clinical benefit compared to inhibition of only BRAF[4,5]. More recently, both clinical[6,7] and pre-clinical[8] studies have shown that inhibition of three compo-nents of the same oncogenic pathway further increases ther-apeutic benefit. In these scenarios the drugs are usually administered at maximum tolerated dose (MTD). The increase in the number of drugs being used in combination is often accompanied by an increase in toxicity and to this date virtually no studies have been done to assess the efficacy of using drugs below-MTD. In a preclinical model, multiple drugs used at low dose also demonstrated promising activity in ovarian clear cell carcinoma[9]. In this study, we explore the use of a Multiple Low Dose (MLD) strategy in *EGFR* mutant NSCLC. In this approach, multiple drugs that act in the same oncogenic signalling pathway are combined at low concentration. We hypothesised that this might add up to complete pathway inhibition without causing prohibitive toxicity. Further, by using low drug concentrations, the pressure exerted on each node of the pathway should greatly diminish, reducing the selective pressure on each node and therefore diminishing the chances of acquiring resistance.

## Results

**MLD therapy blocks MAPK pathway and proliferation in PC9 cells.** The mechanisms of resistance to EGFR inhibition (stan-dard-of-care) in *EGFR* mutant NSCLC are well understood. We therefore compared the efficacy of MLD therapy to standard-of-care MTD therapies in this indication. We used PC9 NSCLC cells, which harbour an activating mutation in the gene encoding EGFR[10]. We used four drugs, each inhibiting a different node in the MAPK pathway: gefitinib (EGFR inhibitor), LY3009120 (pan-RAF inhibitor[11]), trametinib (MEK inhibitor) and SCH772984 (ERK inhibitor[12]), as shown schematically in Fig. 1a. We estab-lished dose-response curves for each of the four drugs using 5-day culture assays (Fig. 1b). From these data, we inferred for all 4 inhibitors the $IC_{20}$ dose, i.e., a drug concentration that inhibits cell viability by 20%—henceforth referred as Low Dose (LD). To assess the efficacy of the MLD strategy we then tested the impact of all possible drug combinations of the 4 drugs at LD on cell viability (assessed by CellTiter-Blue® assay), on cell proliferation (assessed by long-term colony formation assay) and on pathway activity (measured by p-RSK levels[13] using Western Blotting) (Fig. 1c-e). The expected viability and the synergy scores were calculated using the Bliss independence model[14]. We found that PC9 cells treated with the single drugs at low dose were only minimally affected, as expected. However, some of the drug combinations showed a striking combination effect, much higher than expected based on drug additivity. In particular, the com-bination of RAF + MEK + ERK inhibitors at low dose (hence-forth called 3D combination) and the combination of EGFR + RAF + MEK + ERK inhibitors at low dose (henceforth called 4D combination) showed an almost complete inhibition of cell via-bility and proliferation, along with a complete blockade of MAPK pathway signalling. Due to these notable findings we pursued the MLD study focusing on the 3D and 4D combinations. To address

if we could further reduce the drug concentrations, we diluted the 4D combination. When the drugs were reduced to half of the $IC_{20}$ concentrations, the 4D combination was no longer able to achieve complete inhibition of proliferation and was similarly unable to mediate complete MAPK pathway inhibition, indicating that there is a threshold that limits efficacy (Supplementary Fig. 1a, b). Based on this, we continued our MLD studies using the $IC_{20}$ concentrations as "Low Dose". To make sure our findings were not drug-specific, we tested the MLD approach using different inhibitors for each of the nodes in the MAPK pathway (erlotinib as EGFRi, BGB-283 as RAFi, selumetinib as MEKi and LY-3214996 as ERKi). Supplementary Fig. 1c, d show that we obtained essentially the same effect with these drugs in 3D and 4D combinations. This, together with the notion that each drug is used at low dose, makes it very unlikely that off target effects of the four drugs are responsible for the observed effects.

**MLD therapy minimises therapeutic resistance.** Next, we tested how MLD therapy compares to standard-of-care high dose therapy in terms of resistance development. To mimic high dose therapy, we treated PC9 cells with a concentration of EGFR inhibitor gefitinib that inhibited cell viability by ~99% in a 5-day culture assay – henceforth referred as High Dose (HD). We found that 3D and 4D combinations inhibit cell proliferation and induce apoptosis at comparable levels to cells treated with HD of gefi-tinib (Fig. 2a and Supplementary Fig. 2a, b). The level of pathway inhibition is also similar between cells treated with 3D and 4D combinations and HD of gefitinib (Fig. 2d). In addition, we performed RNA-Seq transcriptome analyses in cells treated with 4D combination (Supplementary Fig. 2c, d). These data showed that 4D combo treated cells displayed a significant down-regulation of MYC and E2F target genes, as well as cell cycle genes. Moreover, MAPK activity markers[15] were significantly downregulated and several pro-apoptotic genes were found to be upregulated, while anti-apoptotic genes were downregulated. To study how MLD therapy compares to HD therapy regarding resistance, we treated PC9 cells with 3D or 4D combinations and with HD of gefitinib or osimertinib for one month (Fig. 2b). As seen by others previously[16,17], cells treated with HD of gefitinib or osimertinib quickly developed resistance, but the cells treated with 3D or 4D combinations did not. In addition, we treated PC9 cells for 16 days with high dose of gefitinib or with 3D or with 4D MLD combinations; we then either removed the drugs, continued to treat with the original drug, or treated with 4D MLD combi-nation for another 16 days (Supplementary Fig. 2e). We observed resistant colonies after 32 days of gefitinib treatment, but not in the cells treated with 3D or 4D combinations. Apparently, 16 day-treatment with 3D or 4D combinations had killed all cells, as continued culturing for another 16 days in media without drugs did not yield any colonies. Importantly, PC9 cells that had developed resistance to high dose EGFR inhibitor, were still responsive to 4D MLD combination. This striking result indicates that EGFR inhibitor-resistant cells remain sensitive to 3D and 4D combinations. This suggests that MLD therapy might be an option for patients having developed resistance to standard-of-care EGFR inhibitor therapy.

**MLD therapy is effective in EGFRi-resistant PC9 cells.** To study further if EGFRi-resistant cells are indeed sensitive to 3D and 4D combinations, we generated PC9 cells resistant to clinically-used EGFR inhibitors. We cultured PC9 cells in the presence of gefi-tinib (PC9-GR) or osimertinib (PC9-OR) until cells were no longer responsive to the inhibitors (see methods). We performed exome sequencing of the two resistant cell populations to gain insight into the mechanisms of acquired resistance. These data

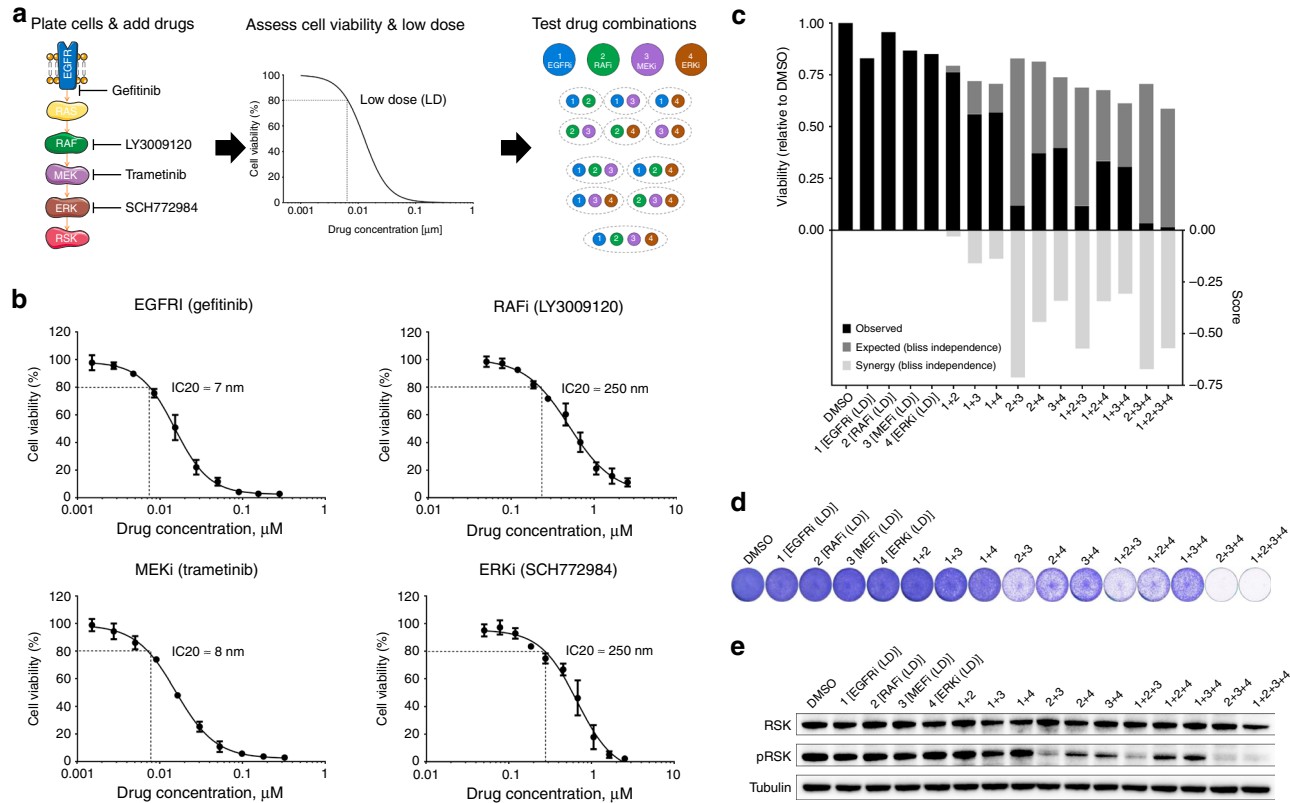

**Fig. 1 Multiple low dose therapy blocks MAPK pathway and proliferation in PC9 cells. a** Schematic of the Multiple Low Dose (MLD) efficacy determination. After plating, cells are treated with increasing drug concentrations. Four days later cell viability is measured and the low dose (LD) is assessed. At last, the efficacy of all the possible combinations at LD is determined. **b** Dose-response curves of EGFR, RAF, MEK and ERK inhibitors in PC9 cells. PC9 cells were cultured with increasing concentrations of EGFRi Gefitinib, RAFi LY3009120, MEKi Trametinib or ERKi SCH772984 for 4 days, after which cell viability was measured using CellTiter-Blue®. Standard deviation (SD) from 3 biologically independent replicates (each with 3 technical replicates) is plotted. Low doses (IC20s) were then determined: gefitinib=7 nM, LY3009120 = 250 nM, 292 trametinib = 8 nM and SCH772984 = 250 nM. **c–e** Determination of the efficacy of all the possible combinations of EGFR, RAF, MEK and ERK inhibitors at LD in PC9 cells. PC9 cells were cultured with all possible drug combinations of EGFR, RAF, MEK and ERK inhibitors at the low doses determined in **b**. In **c** cell viability from 3 biologically independent replicates (each with at least 3 technical replicates) was measured by CellTiter-Blue® assay after 4 days of treatment; In black the observed experimental viability; In dark-grey the expected viability and in light-grey the synergy scores, calculated using the Bliss independence model, are plotted. In **d** cells were treated for 10 days, after which plates were stained and scanned; A representative image from the 3 biologically independent replicates performed is displayed. In **e** protein for western blotting was harvested after 24 h of treatment; The level of pathway inhibition was determined by examining pRSK protein levels in the western blot. Tubulin was used as loading control.

showed acquisition of the well-known T790M mutation in the PC9-GR cells and a number of mutations in the PC9-OR cells, none of which has been previously associated with resistance to osimertinib (Supplementary Table 2). We then tested the sensitivity of the resistant lines to 3D and 4D combinations. In both resistant cell populations, we saw an almost complete inhibition of cell viability after only 4 days of MLD therapy treatment and a complete MAPK pathway signalling blockade (Fig. 2c, d).

**MLD therapy is effective in multiple tumour models**. We then tested if the MLD strategy would also be effective in additional in vitro tumour models. After low dose determination (Supplementary Fig. 3a-c and Supplementary Table 1) we tested the MLD strategy in patient-derived (colorectal and NSCLC) organoids. Treatment with 3D and 4D combinations resulted in a major reduction in cell viability (Fig. 3a). In addition, we tested 6 different MAPK pathway addicted cell lines: HCC827 and H3255 (*EGFR* mutant lung cancer), H2228 and H3122 (*EML4-ALK* translocated lung cancer, in which EGFRi was replaced with ALK inhibitor crizotinib in the 4D combination), DiFi and Lim1215 (EGFR dependent colorectal cancer) and in 2 different PI3K pathway addicted cell lines: SKBR3 and HCC1954 (*HER2*

amplified breast cancer, in which 4D combination consisted of HER2, PI3K, AKT and mTOR inhibitors). When treated with 4D combination, proliferation of all cell lines was inhibited, regardless of the tumour type/driver/genotype, pointing towards a broad applicability of the MLD treatment strategy (Supplementary Fig. 3d).

**MLD therapy is tolerated by non-tumorigenic cell lines**. One of the major concerns when using multiple drugs in combination is the possible toxicity to normal tissues[18]. To test the effect of the MLD strategy on "normal" (non-tumorigenic) cell lines we used primary human BJ (fibroblast) and RPE1 (retinal pigment epithelium) cells. Upon 3D and 4D MLD drug combination treatment, cell viability was reduced, but to a much lesser extent than in cancer cells. This indicates that the MLD strategy might be tolerated by normal tissues (Fig. 3b). Since the MAPK pathway is rich in cross-talk and feedback control circuits[19,20], we also tested how a pulse of signalling through the EGFR pathway would be affected by 3D or 4D MLD treatment. We serum-starved BJ cells overnight and then incubated with 3D or 4D MLD drug combinations for 2 h. After this, cells were stimulated with 100 ng/mL of EGF in the presence of 3D or 4D drug combinations. Twenty

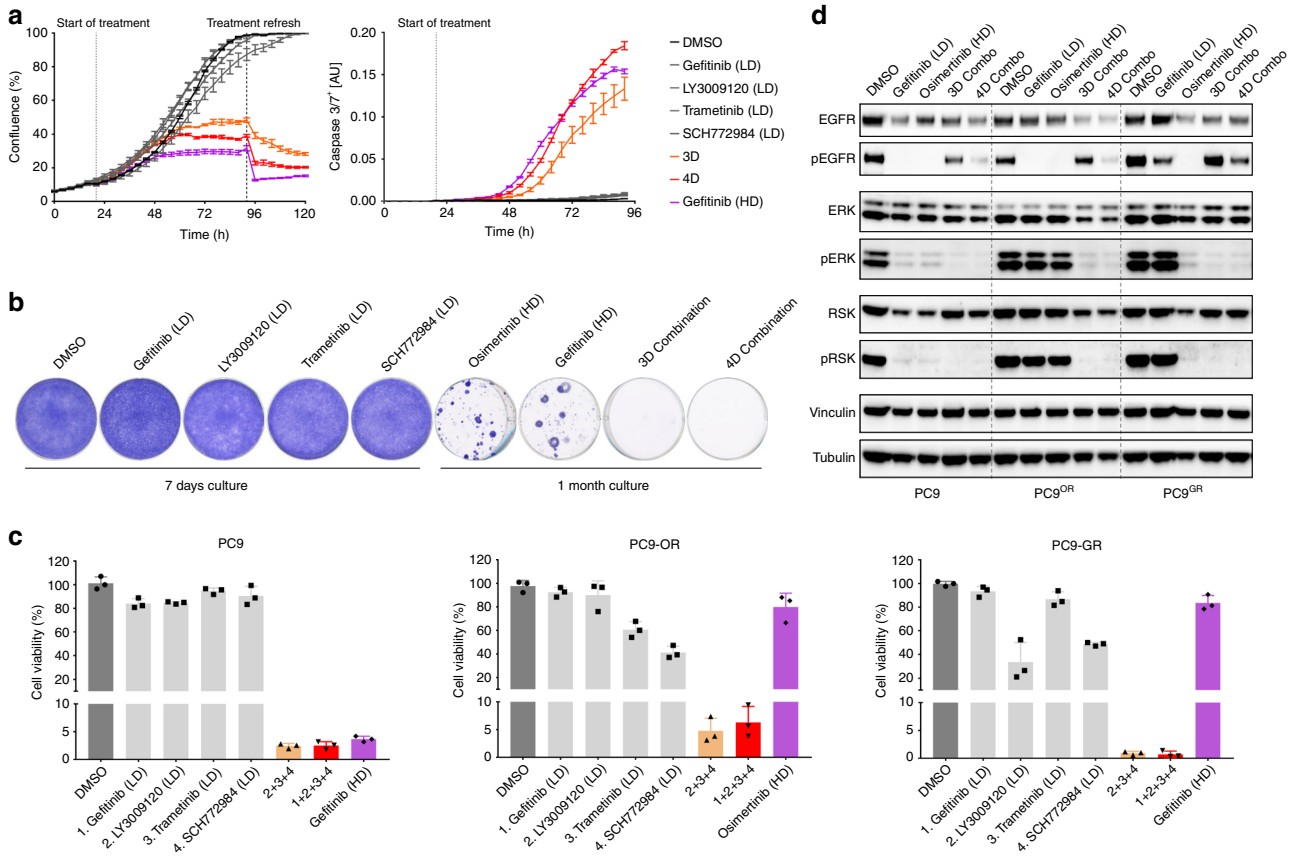

**Fig. 2 MLD therapy minimises therapeutic resistance and is effective in EGFRi-resistant PC9 cells. a** MLD therapy abrogates cell proliferation and induces apoptosis in PC9 cells. PC9 cells were plated and incubated overnight to allow attachment to the plate. Cells were then treated with DMSO, with EGFR, RAF, MEK, ERK inhibitors at low dose, with 3D Combo (RAF + MEK + ERK inhibitors at LD) or with 4D Combo (EGFR + RAF + MEK + ERK inhibitors at LD) and placed in the IncuCyte®. Confluence (left) and caspase 3/7 activation (right) over time was measured by the IncuCyte®. Standard error of the mean (SEM) from 3 replicates is plotted. **b** MLD therapy prevents the acquisition of drug resistance in PC9 cells. PC9 cells were cultured with DMSO, with EGFR, RAF, MEK and ERK inhibitors at low dose (for 7 days) and with high dose (HD) of Osimertinib (200 nM), HD of Gefitinib (280 nM) and with 3D and 4D Combinations (for 1 month), after which plates were stained and scanned; A representative image from 3 biologically independent replicates is displayed. **c** EGFRi-resistant PC9 cells remain sensitive to MLD therapy. PC9, PC9-OR (Osimertinib-resistant) and PC9-GR (Gefitinib-resistant) cells (see methods) were cultured with DMSO, with low doses of EGFR, RAF, MEK or ERK inhibitors, with 3D or 4D combinations or with HD of Gefitinib or Osimertinib for 4 days, after which cell viability was measured using CellTiter-Blue®. Standard deviation (SD) from 3 biologically independent replicates is plotted. **d** MLD therapy blocks MAPK pathway in EGFRi-resistant PC9 cells. PC9, PC9-OR and PC9-GR cells were cultured with DMSO, HD of Osimertinib, HD of Gefitinib or with 3D or 4D combinations. Protein for western blotting was harvested after 24 h of treatment; The level of pathway inhibition was measured by examining pERK and pRSK protein levels and the level of EGFR inhibition was measured by examining pEGFR protein levels in the western blot. Tubulin and Vinculin were used as loading control.

minutes after EGF stimulation, a significant amount of p-RSK was detected, which was no longer detected at 4 h post EGF stimulation (Fig. 4c). These data suggest that the efficient inhibition of MAPK signalling exerted by 3D and 4D MLD treatment is the result of an effect of these drugs on homoeostatic feedback/cross-talk signalling[19,21], as pulsatile signalling through the MAPK pathway seems to be much less affected than persistent signalling through an oncogene-activated MAPK pathway.

**MLD therapy induces tumour regression without toxicity in vivo.** To address if the MLD strategy is effective in vivo, we used patient derived xenograft (PDX) tumours from four different patients who had developed resistance to first-line or second-line therapy with EGFR inhibitors erlotinib or osimertinib[22] in the clinic by acquiring EGFR T790M mutation, *KRAS* mutation or *MET* amplification (Supplementary Table 3). For the in vivo studies we defined LD as 20% (for gefitinib and trametinib) and 30% (for LY3009120 and SCH772984—due to the shorter half-

lives) of the published maximum tolerated dose (MTD) in mice for each of the individual drugs[11,12,23,24]. Osimertinib-resistant PDX-1 was implanted subcutaneously and orthotopically in the lungs. In both models, treatment with 3D or 4D combination resulted in a reduction in tumour volume, without associated toxicity (Fig. 4a–d). Interestingly however, treatment with 4D combo was slightly more effective than 3D combo. Due to this finding we focused the in vivo studies that followed on the 4D combination. In all PDX models tested we observed similar results to PDX-1, i.e., a reduction in tumour volume, without significant toxicity (Fig. 4e, f and Supplementary Fig. 4e). In addition, in gefitinib-resistant models PDX-2 and PDX-3 we tested if it would be possible to acquire resistance to the 4D MLD combination therapy during a drug holiday. In both PDX models, re-starting of 4D MLD therapy after a drug holiday resulted in a second response to the drug combination, indicating that overt resistance had not developed in vivo (Fig. 4e, f).

We also implanted PC9 cells in nude mice and treated them with vehicle, with EGFR, RAF, MEK and ERK inhibitors

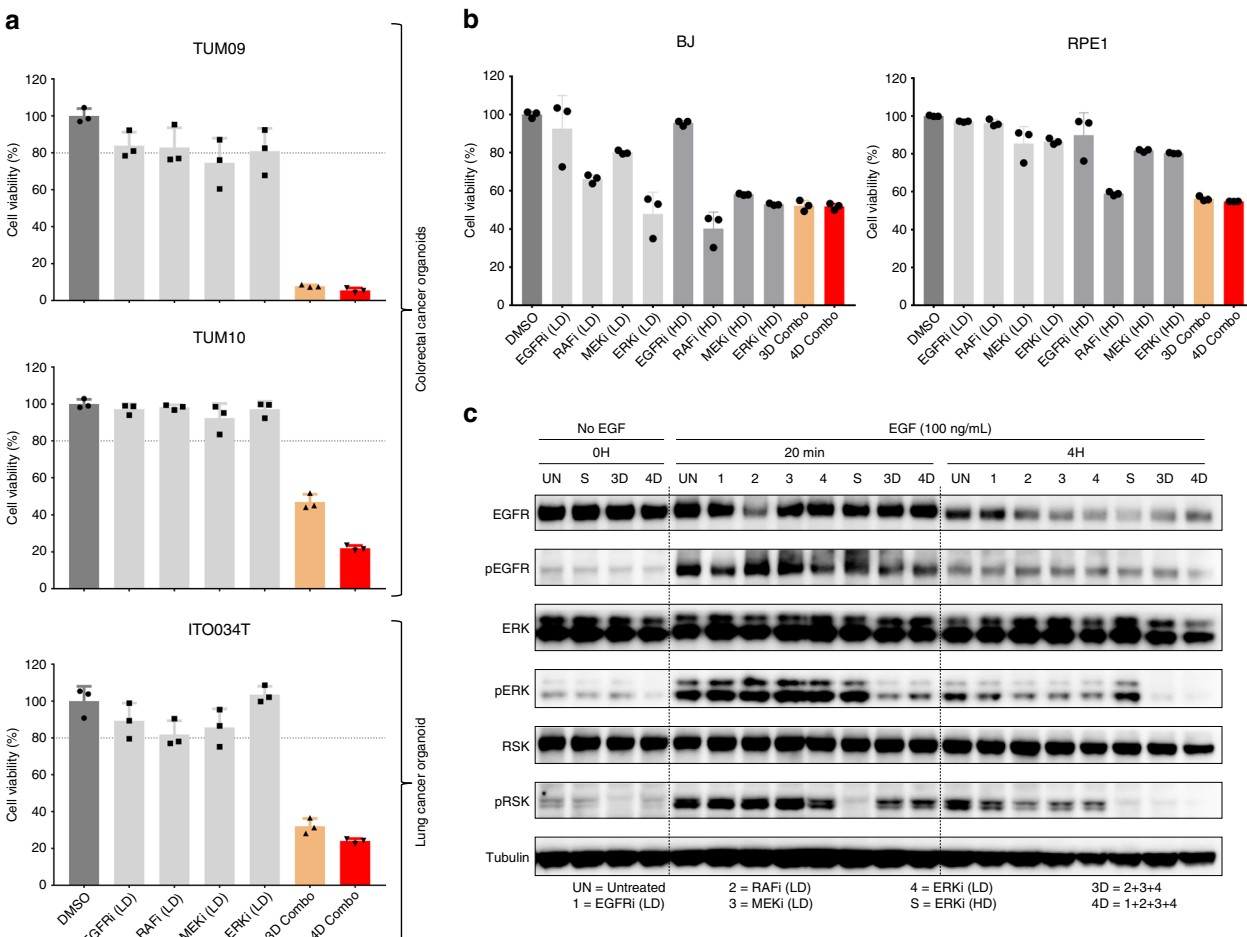

**Fig. 3 MLD therapy is effective in patient-derived organoids and is tolerated by normal cell lines. a** MLD therapy is effective in several colorectal and lung cancer patient-derived organoids. Organoids were cultured with DMSO, with EGFR, RAF, MEK and ERK inhibitors at LD and with 3D and 4D combos. After 5 days of drug treatment cell viability was measured using CellTiter-Glo®. Standard deviation (SD) from 3 biologically independent replicates is plotted. **b** Cell viability of normal cells is much less affected by MLD therapy than tumour cells. BJ and RPE1 cells were treated with DMSO, with EGFR, RAF, MEK and ERK inhibitors at low and high doses and with 3D and 4D Combos (using the LD and HD concentrations determined for PC9 cells). After 4 days of drug treatment cell viability was measured. SD from 3 replicates is plotted. **c** MLD therapy allows pulsed signaling in normal cells. BJ cells, after overnight starvation, were treated with the indicated inhibitors/concentrations for 2 h, after which EGF (100 ng/mL) was added. Cells were harvested before, 20 min and 4 h after EGF stimulation. The level of pathway inhibition was measured by examining pERK and pRSK protein levels. The level of EGFR inhibition was measured by examining pEGFR protein levels in the western blot. Tubulin was used as loading control.

individually at low dose and with 4D combination. The use of low dose regimens was inadequate to suppress PC9 tumour growth when used as single agents, but when used in combination we observed a sustained reduction in the tumour volume of PC9 xenografts over a period of 70 days, which was associated with an extended survival (Supplementary Fig. 4a, b). These observations are also supported by immunohistochemical staining of the tumours, which show decreased Ki67 (a proliferation marker) and pERK (MAPK activation) levels in the tumours treated with 4D combination (Fig. 4g). Significantly, mice treated with 4D combination did not show any significant signs of toxicity, assessed by the weight of the mice over time and by the morphology of the GI tract and bone marrow (Supplementary Figs. 4c, f). In the clinic, the T790M mutation is already present (at very low percentages) in the majority of the tumours before undergoing anti-EGFR treatment[25,26]. To mimic this scenario, we implanted in nude mice a mix of PC9 cells and PC9-GR cells (which are T790M positive) in a 9:1 ratio, respectively. Mice were treated with vehicle, with MTD of gefitinib and with 4D combination. Treatment with MTD of gefitinib resulted in a quick reduction of tumour volume which was followed by

outgrowth of resistant cells, unlike the mice treated with 4D combination, where a sustained tumour control was observed (Supplementary Fig. 4d).

Despite the significant tumour regressions observed in the in vivo experiments none of the mice were fully cured, unlike in the in vitro data where all the cells were killed by the 3D or 4D combinations. To study why this is the case we studied the pharmacokinetics and pharmacodynamics of the four drugs in vivo over time. We found that drug plasma concentrations of gefitinib and trametinib dropped relatively slowly ($T_{1/2}$ 8 h), but the pan-RAF and ERK inhibitors were less stable in plasma ($T_{1/2}$ of 5 and 4 h, respectively). A similar difference was seen for intra-tumoural drug concentrations (Supplementary Figs. 5a, b). Consistent with this, we observed a complete inhibition of pRSK in tumour biopsies 2 h after 4D combination drug administration, which progressively decreased after 8 and 24 h (Supplementary Fig. 5c). These data indicate that, unlike in the in vitro experiments, two of the four drugs were not present at a significant concentration during at least 12 h of the 24-h treatment cycle. As a result of this, a sustained MAPK pathway inhibition was not achieved in vivo, possibly explaining why we

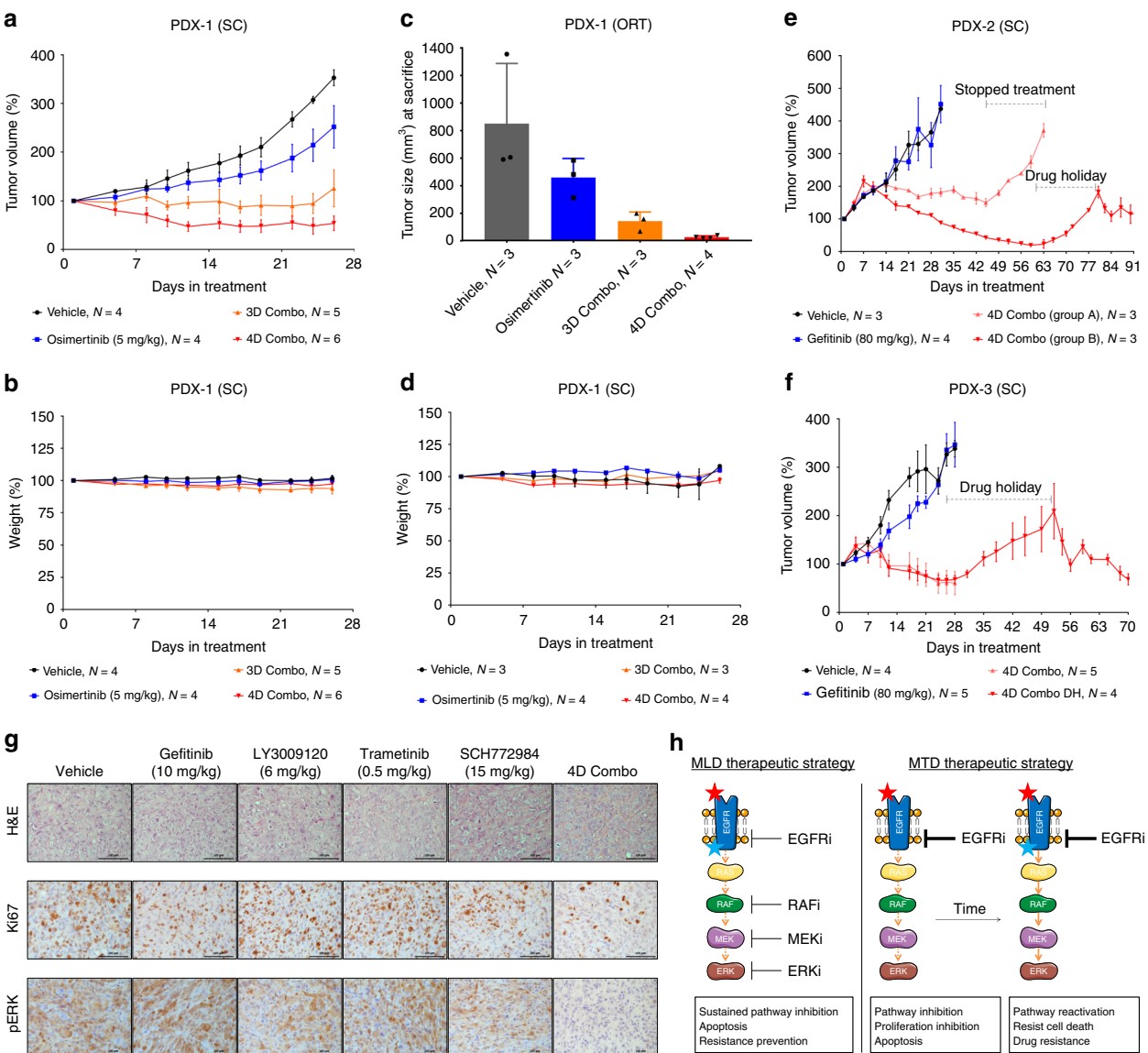

**Fig. 4 MLD therapy induces tumour regression without toxicity in vivo. a–f** Patient derived xenografts (PDX) are sensitive to MLD therapy. PDX tumours (see Supplementary Table 3) were implanted subcutaneously (**a**, **e**, **f**) or orthotopically in the lungs (**c**) of Crl:NU-Foxn1nu mice. PDX1 was implanted both subcutaneously (**a**) and orthotopically in the lungs (**c**). We defined the in vivo LD as 20–30% of the MTD for each of the individual drugs–gefitinib (10 mg/kg), LY3009120 (6 mg/kg), trametinib (0.5 mg/kg) and SCH772984 (15 mg/kg). In **a–d** after tumour establishment, mice were treated 5 days/week with vehicle, with osimertinib (5 mg/kg) and with 3D or with 4D Combos for 26 days, after which mice were sacrificed. In **a** tumour volume percentages ±SEM is shown, in **c** tumour size (mm³) at sacrifice ±SEM is shown and in **b** and **d** the mice weight percentages ± SEM is shown. **e** After tumour establishment, mice were treated 5 days/week with Vehicle ($N=3$), with gefitinib (80 mg/kg) ($N=4$) or with 4D Combo for 6 weeks (group A, $N=3$) or with 4D Combo for 8 weeks (group B, $N=3$). Mice treated with vehicle and gefitinib were sacrificed when tumours reached ~2000 mm³. After 6 weeks, Group A was taken off treatment and mice were sacrificed when tumours reached ~2000 mm³. After 8 weeks Group B was taken off treatment and was given 3 weeks of drug holiday. Mice were then treated for another 2 weeks with 4D combo, after which they were sacrificed. Tumour volume percentages ±SEM is shown. **f** After tumour establishment, mice were treated 5 days/week with vehicle ($N=4$), with gefitinib (80 mg/kg) ($N=5$) or with 4D Combo ($N=9$) for 4 weeks, after which mice were sacrificed, except for 4 animals from the 4D Combo group. These 4 mice were spared and were given 3 weeks drug holiday (4D Combo DH group), followed by another 3 weeks of treatment, after which they were sacrificed. Tumour volume percentages ±SEM is shown. **g** H&E, Ki67 and pERK stainings from tumour sections of PC9 xenografts. A representative staining image from each cohort ($N \geq 6$) is displayed. Scale bars 100 µm. **h** Schematic representation of the MLD therapy for the treatment of EGFR mutant NSCLC.

didn't achieve full tumour regressions. We tested this hypothesis in vitro, by removing RAF and ERK inhibitors from the treatment for approximately 8 h every day. We found that, as hypothesised, when the drugs in the 3D or 4D combination were not present continuously the MLD therapy became less effective (Supplementary Fig. 5d). Finally, we tested whether there was any drug-drug interaction by measuring the half-lives of the drugs when given alone or in combination (Supplementary Fig. 5e). Overall,

there is not any apparent drug-drug interaction, except for LY3009120, for which the half-life increases when given in the 4D Combo.

## Discussion

We report here that treatment of *EGFR* mutant NSCLCs with MLD therapy effectively suppresses development of drug

resistance, without associated toxicity. As such, our data challenge the common paradigm that patients should be treated with the MTD of a targeted agent. Our data are consistent with a model in which diffuse inhibition of an oncogenic pathway at multiple nodes reduces selective pressure on each of the nodes to mutate and thereby increase response time (Fig. 4h). Our findings also challenge the current model for MAPK pathway signalling, which postulates that the MAPK kinase cascade functions to amplify signals. Such amplification cascade model is clearly at odds with the data obtained here in which a very partial inhibition of each of 4 nodes in this cascade adds up to complete pathway inhibition. Further mechanistic studies are required to better understand the efficacy of the MLD strategy.

Importantly, we show that tumours having the most common mechanisms of clinically-observed resistance to high dose standard of care EGFR inhibitors still respond to MLD therapy. Therefore, MLD treatment strategy appears especially promising for patients that have already developed resistance to all clinically used EGFR inhibitors, including osimertinib. In such resistant tumours, multiple metastases may be present having different resistance mutations. In this study we have shown that MLD therapy is effective in PDX models having diverse EGFR inhibitor resistance mechanisms, including EGFR T790M mutation, *KRAS* mutation, *MET* amplification and even SCLC transformation, highlighting that MLD therapy could apply to a diverse range of EGFR TKI resistant tumours. However, not all the resistance mechanisms have been tested and it is possible that some might not respond to MLD therapy. Indeed, in clinical practice, an MLD treatment strategy can only be tested in patients having developed resistance to standard-of-care EGFR inhibitors. We find in PDX models that 4D MLD is consistently somewhat better than 3D MLD, which may relate to the notion that not all *EGFR* alleles in the tumour may have acquired resistance mutations to the EGFR inhibitor therapy. Furthermore, it will be important to maintain osimertinib in an experimental MLD therapy trial, as this drug crosses the blood-brain barrier, and such late-stage patients may have (latent) brain metastases. We therefore suggest that clinical testing of the MLD strategy should include osimertinib.

While we never observed development of resistance to MLD therapy in vivo, even after long drug exposure, we did not achieve complete tumour regressions. This is most likely due to the short half-lives of the RAF and ERK inhibitors used in this study, which resulted in a situation in which we did not achieve a continuous pathway blockade. This may be improved by using continuous release formulations of these drugs, or by using drugs with longer half-lives.

The MLD therapy described here is fundamentally different from metronomic chemotherapy[27,28]. In this latter scenario, low doses of chemotherapy are given at high frequency with the aim to suppress division of endothelial cells of the tumour vasculature. In the present MLD schedule, we target the MAPK pathway of the tumour itself, as growth inhibition in all cases parallels inhibition of the MAPK pathway (as judged by pRSK). Three-drug combinations given at MTD have been used before in pre-clinical[8] and clinical studies[6,7] for *BRAF^V600E* mutant tumours, showing clear therapeutic benefits, but such regimen have an associated cost of toxicity.

The lack of significant toxicity of the MLD therapy in mice may be explained by the fundamentally different nature of MAPK pathway signalling between normal and EGFR mutant cancer cells. In the former, signalling is transiently activated when growth factors are present. In the latter, oncogenic mutations result in persistent activation of the pathway. Importantly, we show here that transient signalling in normal cells is, at least initially, not interrupted by MLD treatment (Fig. 3c). This may explain why long-term exposure of mice to MLD treatment is

without major toxicity, as judged by lack of weight loss and lack of toxicity to gut epithelium and bone marrow. However, mice and human are fundamentally different with respect to drug toxicity. Especially skin toxicity following MAPK inhibition therapy is often underestimated in mice. Therefore, only a phase I clinical trial will be able to fully assess the toxicity of this strategy in humans.

Extrapolation of dose from animals to humans based only on mg/kg conversion is difficult, since body surface area and differences in pharmacokinetics should also be taken into consideration. To convert the animal dose in mg/kg to human equivalent doses (HED) in mg/kg, it is recommended to divide by 12.3[29]. If we estimate the HED based on the low-doses used in our in vivo experiments for Gefitinib and Trametinib (where dosing in humans is known) using this approach then Gefitinib (10 mg/kg in mice) corresponds to 57 mg once daily in patients, which is approximately one quarter of the dose used in patients (250 mg qd). And Trametinib (0.5 mg/kg in mice) corresponds to 2,8 mg once daily in patients, which is a bit higher than the dose used in patients (2 mg qd). However, we also we performed an in vivo experiment using lower concentrations of Gefitinib (1 mg/kg) and Trametinib (0.1 mg/kg) (Supplementary Fig. 4d). These drug concentrations correspond to 2,5% of the human daily dose for gefitinib and 28% of the daily human dose of trametinib, using the calculation method of Nair mentioned above. These data indicate that with these further reduced concentrations of Gefitinib and Trametinib we still have a significant anti-tumor effect in vivo. Due to the difficulty in translating drug doses from mice to human we feel that only a well-designed phase 1 trial can help assess the potential clinical utility of the MLD strategy proposed here.

Even though we focused mostly on EGFR mutant NSCLC, we have also shown that the MLD strategy can potentially be effective in other tumour types. Overall, our findings challenge the current paradigm of using the maximum tolerated dose of single targeted cancer drugs and suggest that, instead, it might be more beneficial to use a combination of multiple drugs that target the oncogenically activated pathway using sub-optimal drug concentrations.

## Methods

**Cell lines culture and drug-response assays.** The PC9 cell line was obtained from ATCC. PC9^OR (osimertinib-resistant) and PC9^GR (gefitinib-resistant) cells were made by continuous (2 months) drug exposure of PC9 cells to 1 μM osimertinib (AZD9291) and to 2 μM gefitinib, respectively. Exome sequencing was performed to determine if any de novo genetic alterations had occurred (Supplementary Table 2). The HCC827, H3255, H3122, H2228, SKBR3, HCC1954, BJ and RPE1 cell lines were obtained from ATCC. And DiFi and Lim1215 cell lines were a gift from A. Bardelli (Torino, Italy). BJ and RPE1 cells were cultured in DMEM (Gibco 41966029). SKBR3 and HCC1954 which were cultured in DMEM/F-12 medium (Gibco 31331028). All the other cell lines were cultured in RPMI medium (Gibco 21875034). All the cell lines media were supplemented with 10% FBS (Serana), 1% penicillin/streptomycin (Gibco 15140122) and 2 mM L-glutamine (Gibco 25030024). All cell lines were cultured at 37 °C and with 5% $CO_2$. All cell lines were validated by STR profiling and mycoplasma tests were performed every 2–3 months.

All drug-response assays were performed in triplicate, using black-walled 384-well plates (Greiner 781091). Cells were plated at the optimal seeding density (Supplementary Table 1) and incubated for approximately 24 h to allow attachment to the plate. Drugs were then added to the plates using the Tecan D300e digital dispenser. 10 μM phenylarsine oxide was used as positive control (0% cell viability) and DMSO was used as negative control (100% cell viability). Four days later, culture medium was removed and CellTiter-Blue (Promega G8081) was added to the plates. After 1–4 h incubation, measurements were performed according to manufacturer's instructions using the EnVision (Perkin Elmer).

**Organoid culture and drug-response assays.** Colorectal (CRC) and non-small cell lung cancer (NSCLC) organoids were established and handled as previously described[30]. All drug-response assays were performed in replicate, each by independent researchers. PDOs were mechanically and enzymatically dissociated into single cells, pipetted through a 40 μM cell strainer, and re-plated to allow for

organoids formation. At day 4 PDOs were collected, Cultrex was removed by incubation of the cell pellet in 1 mg/mL dispase II (Sigma D4693) for 15 min. Whole organoids were counted using a hemocytometer and trypan blue. PDOs were resuspended in 1:3 Advanced Dulbecco's Modified Eagles Medium with Nutrient Mixture F-12 Hams (Ad-DF) (Invitrogen 12634), supplemented with 1% penicillin/streptomycin (Invitrogen 15140122), 1% HEPES (Invitrogen 15630056) and 1% GlutaMAX (Invitrogen 35050) (Ad-DF+++):Cultrex at a concentration of 20 organoids/μL. Five μl/well was dispensed in clear-bottomed, white-walled 96-well plates (Greiner Bio-One 655098) and overlaid with 200 μL CRC or NSCLC culture medium. We generated 10-step dose response curves using the Tecan D300e digital dispenser, interpolated $IC_{20}$ values and re-screened organoids in presence of a range of concentration around the $IC_{20}$ of each drug separately and in 3D and 4D Combos. In addition, we re-performed the dose-response curves to control for variation between experiments. Read-out was performed at day 10 in the positive control (10 μM phenylarsine oxide), negative control (DMSO), and the drug-treated wells. Quantification of cell viability was done by replacing the CRC medium with 50 μL Cell-TiterGlo 3D (Promega G9681) mixed with 50 μL Ad-F+++. Measurements were performed according to manufacturer's instructions on an Infinite 200 Pro plate reader (Tecan Life Sciences) with an integration time of 100 ms.

### Compounds, reagents and antibodies.

Gefitinib (100140), LY3009120 (206161), trametinib (201458), SCH772984 (406578), osimertinib (206426), crizotinib (202222), lapatinib (100946), BKM120 (204690), MK2206 (201913) and AZD8055 (200312) were purchased from MedKoo Biosciences. Erlotinib (S7786), BGB-283 (S7926), selumetinib (S1008) and LY-3214996 (S8534) were purchased from Selleckchem. Annexin V-FITC Apoptosis Staining Detection Kit was purchased from Abcam (ab14085).

Antibodies against Tubulin (T9026) and Vinculin (V9131) were purchased from Sigma; antibodies against EGFR (4267), pERK (4377), ERK (9102) and RSK (8408) were purchased from Cell Signalling; antibody against pRSK (04-419) was purchased from Millipore; antibody against pEGFR (ab5644) was purchased from Abcam. Secondary antibodies Goat Anti-Rabbit IgG (H + L)-HRP Conjugate (1706515) and Goat Anti-Mouse IgG (H + L)-HRP Conjugate (1706516) were purchased from Bio Rad.

### Colony formation and IncuCyte cell proliferation assays.

Cells were seeded in the appropriate density (Supplementary Table 1) in 6-well plates. Cells were incubated for approximately 24 h to allow attachment to the plates, after which drugs were added to the cells using the Tecan D300e digital dispenser as indicated. The culture media/drugs were refreshed every 3/4 days. When control wells (DMSO) were confluent (unless otherwise stated in the text) cells were fixed using a solution of 2% formaldehyde (Millipore 104002) diluted in phosphate-buffered saline (PBS). Two hours later, they were stained, using a solution of 0.1% crystal violet (Sigma HT90132) diluted in water. Not more than 10 min later the staining solution was removed, plates were washed with water left to dry overnight. Finally, plates were scanned and stored.

For IncuCyte proliferation assays, cells were seeded in 96-well plates and incubated overnight to allow attachment to the plates. Drugs were added to the cells using the Tecan D300e digital dispenser. Cells were imaged every 4 h in the IncuCyte ZOOM (Essen Bioscience). Phase-contrast images were collected and analysed to detect cell proliferation based on cell confluence. For cell apoptosis, IncuCyte® Caspase-3/7 green apoptosis assay reagent (Essen Bioscience 4440) was also added to culture medium and cell apoptosis was analysed based on green fluorescent staining of apoptotic cells.

### Western Blots.

After the indicated culture period, cells were washed with chilled PBS and then lysed with RIPA buffer (25 mM Tris-HCl pH 7.6, 150 mM NaCl, 1% NP-40, 1% sodium deoxycholate, 0.1% SDS) containing protease inhibitors (Complete (Roche) and phosphatase inhibitor cocktails II and III). Samples were then centrifuged for 10 min at 14.000 rpm at 4 °C and supernatant was collected. Protein concentration of the samples was normalised after performing a Bicinchoninic Acid (BCA) assay (Pierce BCA, Thermo Scientific), according to the manufacturer's instructions.

Protein samples (denatured with DTT followed by 5 min heating at 95 °C) were then loaded in a 4–12% polyacrylamide gel. Gels were run (SDS-PAGE) for approximately 60 min at 165 V. Proteins were then transferred from the gel to a polyvinylidene fluoride (PVDF) membrane, using 330 mA for 90 min.

After the transfer, membranes were placed in blocking solution (5% bovine serum albumin (BSA) in PBS with 0,1% Tween-20 (PBS-T). Subsequently, membranes were probed with primary antibody in blocking solution (1:1000) and left shaking overnight at 4 °C. Membranes were then washed 3 times for 10 min with PBS-T, followed by 1 h incubation at room temperature with the secondary antibody (HRP conjugated, 1:10000) in blocking solution. Membranes were again washed 3 times for 10 min in PBS-T. Finally, a chemiluminescence substrate (ECL, Bio-Rad) was added to the membranes and the Western Blot was resolved using the ChemiDoc (Bio-Rad).

### Mouse xenografts studies.

All animal experiments were approved by the Animal Ethics Committee of the Netherlands Cancer Institute or by the Animal Ethics Committee of the Institut Català d'Oncologia and performed in accordance with institutional, national and European guidelines for Animal Care and Use.

PC9 cell line xenografts: One million PC9 cells were resuspended in PBS and mixed 1:1 with matrigel (Corning 354230). Cells were injected subcutaneously into the posterior flanks of 7-week-old immunodeficient BALB/cAnNRj-Foxn1nu mice (half male and half female; Janvier Laboratories, The Netherlands). Tumour formation was monitored twice a week. Tumour volume, based on calliper measurements, was calculated by the modified ellipsoidal formula (tumour volume = $1/2$(length × width$^2$)). When tumours reached a volume of approximately 200 mm$^3$, mice were randomized into the indicated treatment arms. Vehicle, gefitinib, LY3009120, trametinib, SCH772984 or the combination of the 4 inhibitors were formulated in DMSO: Kolliphor EL (Sigma 27963): Saline solution, in a ratio of (1:1:8). Mice were treated 5 days a week (Monday to Friday) at the indicated doses by intraperitoneal injection.

Patient-derived xenografts (PDX) and orthotopic xenograft (PDOX): Primary tumours were obtained from Bellvitge Hospital (HUB) and the Catalan Institute of Oncology (ICO) with approval by the Ethical Committee. Ethical and legal protection guidelines of human subjects, including informed consent from the patient to implant the tumour in mice, were followed. PDX-1 was generated from a lung adenocarcinoma biopsy from a patient who was treated with Erlotinib (first line), Gefitinib + Capmatinib (second line) and Cisplatin+Pemetrexed (third line). This tumour has an EGFR mutation (del19) and MET amplification. PDX-2 was generated from a lung adenocarcinoma biopsy from a patient who was treated with Erlotinib (first line), Gefitinib + Capmatinib (second line) and Carboplatin +Gemcitabine and Nivolumab (third line). This tumour has an EGFR mutation (L858R) and MET amplification. PDX-3[30] was generated from a lung adenocarcinoma biopsy of a brain metastasis from a patient who was treated with Erlotinib (first line) and Osimertinib (second line). PDX-4 was generated from a lung adenocarcinoma biopsy from a patient who was treated with Afatinib (first line) and CBDCA + pemetrexed (second line). This tumour has a germline p53 mutation and an EGFR mutation (del19). Tumours were isolated and implanted subcutaneously (or orthotopically, in the lungs, in the case of PDX-3) into Crl:NU-Foxn1nu mice by following previously reported procedures[22,31]. In the subcutaneous models, tumour volume was monitored twice a week by a digital caliper. When tumours reached a volume of approximately 200–600 mm$^3$, mice were randomized into the indicated treatment arms. In the orthotopic model, tumours were left to grow for 2 weeks, followed by 26 days of treatment. Vehicle, gefitinib, osimertinib or the 3D and 4D Combos were formulated in DMSO: Kolliphor EL (Sigma 27963): Saline solution, in a ratio of (1:1:8). Mice were treated 5 days a week (Monday to Friday) at the indicated doses by intraperitoneal injection.

### In vivo pharmacokinetics and pharmacodynamics studies.

Plasma and tumour samples were assayed by liquid chromatography triple quadrupole mass spectrometry (LC-MS/MS) using an API4000 detector (Sciex) for the simultaneous determination of Gefitinib (MRM: 447.4/128.1), LY3009120 MRM: 425.5/324.2), Trametinib (616.3/491.2) and SCH772984 (MRM: 588.4/320.2). Gefitinib-d8 (MRM: 455.4/136.3) was used as internal standard. LC separation was achieved using a Zorbax Extend C18 column (100 × 2.0 mm; ID). Mobile phase A and B comprised 0.1% formic acid in water and methanol, respectively. The flow rate was 0.4 ml/min and a linear gradient from 20%B to 95%B in 2.5 min, followed by 95%B for 2 min, followed by re-equilibration at 20%B for 10 min was used for elution. Sample pre-treatment was accomplished by mixing 5 μL (plasma) or 25 μL (tumour homogenate) with 30 or 150 μL, respectively, of formic acid in acetonitrile (1 + 99) containing the internal standard. After centrifugation, the clear supernatant was diluted 1 + 4 with water and 50 μL was injected into the LC-MS/MS system.

The plasma/tumour samples were harvested at the time points indicated in Supplementary fig. 5. Blood samples were obtained by tail cut (at 2 h and 8 h time points) and by cardiac puncture at the 24 h time point. Samples were collected on ice in tubes containing potassium EDTA as anticoagulant. The tubes were immediately cooled in melting ice and centrifuged (10 min, 5000×g, 4 °C) to separate the plasma fraction, which was transferred into clean vials. For the tumours samples, the mice were sacrificed by cervical dislocation, the tumour was dissected and frozen at −80 °C. Half of the tumour was then lysed mechanically with RIPA buffer and lysates were analysed by Western blot. The other half was weighed and homogenised in 1 mL of ice-cold 1% of BSA in water and stored at −20 °C until further analysis.

### Reporting summary.

Further information on research design is available in the Nature Research Reporting Summary linked to this article.

## Data availability

All relevant data are available from the corresponding author upon request. Full scans of the gels and blots are available in Supplementary Fig. 7. All the other data supporting the findings of this study are available within the article and its supplementary information files. A reporting summary for this article is available as a Supplementary Information file. RNAseq data can be accessed with the GEO assession GSE144258. Most raw data can be assessed at https://doi.org/10.6084/m9.figshare.12408803.v1

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

## Acknowledgements

We thank A. Bardelli (Torino, Italy) for gift of cell lines, the facilities of The Netherlands Cancer Institute: Mouse Clinic–Intervention Unice (Natalie Proost, Bjorn Siteur, Bas van Gerwen, Charlotte Baak, Renske Grimmerink, Rebecca Theeuwsen and Marieke van de Ven), Robotics and Screening Center (Ben Morris and Roderick Beijersbergen), Clinical Pharmacology (Levi Buil and Artur Burylo), Experimental Animal Pathology, Flow Cytometry and Sequencing. We also thank Richard Marais for discussions. This work was supported by a grant from the Dutch Cancer Society through the Oncode Institute. Al.V. was supported by the Fondo de Investigaciones Sanitarias, FIS (PI16-01898, and by the Spanish Association Against Cancer, AECC (CGB14142035THOM) and Ideas Semilla project (IDEAS098VILL-IDEAS16) and Generalitat de Catalunya (2014SGR364). L.F. received a European Union's Horizon 2020 research and innovation programme under the Marie Sklodowska-Curie, grant agreement number 799850. E.N. was funded by Instituto Carlos III through the project PI18/00920. We thank CERCA Program/Generalitat de Catalunya for their institutional support and grant 2017SGR448.

## Author contributions

R.B. supervised all research. J.N., Al.V., O.v.T. and R.B., wrote the manuscript. J.N., E.N., S.O., E.B., L.F., C.M., S.K., A.J., H.H., L.W. and Au.V. designed, performed and analysed the experiments. E.V., Al.V., E.F., A.M.-M. and Lo.W. provided advice for the project. All authors commented on the manuscript.

## Competing interests

Al.V. is co-founder of Xenopat S.L. The remaining authors declare no competing interests.
