## [Peer Review File · Nature Communications]

Reviewers' Comments:

Reviewer #1:

Remarks to the Author:

The paper by Fernandes Neto et al. illustrates a novel approach to treatment of oncogene-addicted cancer based on the combination of multiple signal transduction inhibitors at low-dose. In particular, this MLD approach showed to be effective in preventing and treating resistance to EGFR tyrosine kinase inhibitors in NSCLC cell lines. Preliminary data of possible activity of similar combinations in other experimental models are also provided.

The paper offers novel information that, if validated in a clinical trial, might improve significantly the therapeutic approach in NSCLC patients carrying EGFR mutations. However, there are points that the authors need to clear out in order to improve their manuscript:

1) The authors show in Fig 1C that the combination of different inhibitors has an higher effect than expected based on drug additivity. However, there are specific tools to calculate the additive or synergistic effects of drug combinations that should be used for each of the tested combinations in order to prove that a synergism occurred.

2) The authors affirm that target based agents are used in the clinic at their MTD. However, this is not true for all drugs. For example, the MTD of gefitinib, a drug used in this study, was not reached in phase I clinical trials at a dose of 650 mg. In contrast, the recommended dose of gefitinib for EGFR-mutant NSCLC patients is 250 mg. Therefore, this statement must be changed and it should reflect the current standard clinical practice.

3) The authors need to describe in more details how the PC9 resistant cells were generated. Previous studies have demonstrated that by using different strategies (increasing doses of TKIs, fixed dose of TKI at the IC50 etc...), different mechanisms of resistance are developed by NSCLC cell lines. In addition, some resistant cell lines carry more than one resistance mechanism. Therefore, a more detailed description of the resistant cell lines might add insights to the interpretation of the results.

4) The data presented in figure 3 are not sufficient to affirm that the combination is not toxic for non-transformed cells. ERK inhibitors have been found to be quite toxic in clinical trials. The concentration of inhibitors in the combination is low. In addition, combinations have been shown to be less toxic as compared with single agents in some cases. However, only a phase I clinical trial will be able to assess the toxicity of this combination that cannot be predicted by an in vitro test. This should be clearly stated by the authors.

5) The authors show that the low dose combination targeting the EGFR/MAPK signaling cascade is active in PDX models that carry different mechanisms of resistance and , therefore, they conclude that it might be active in all resistant tumors. However, a fraction of EGFR mutant NSCLC have been shown to become resistant to EGFR TKIs by complex mechanisms leading to transformation in SCLC or epithelial-to-mesenchymal transition (EMT). Unless the authors have the possibility to demonstrate that the 4 drugs combination is active in cells carrying such resistance mechanisms, the limits of their approach should be clearly disclosed in the discussion of the results.

6) The results of the in vivo data are in line with the in vitro data. However, I am not sure that these findings really support the hypothesis that multiple low doses are sufficient for tumor growth inhibition. The authors used doses of drugs corresponding to 20-30% of the MTD for the in vivo studies, although they did not define the specific dose for each drug. If they used 30% of the gefitinib MTD of 10 mg/kg, this corresponds to 3 mg/kg. It means 210 mg for a patient with a weight of 70 kg, which is almost the fixed dose used in the clinic (250 mg). For trametinib, 20% of 0.5 mg/kg is 0.1 mg/kg, i.e 7 mg in a 70 kg patient while the maximum fixed dose approved for treatment of patients is 2 mg. It is evident that the pharmacokinetic and pharmacodynamics of these drugs in mice must be quite different as compared with human. On the other hand, this observation raises several doubts on the possibility to extrapolate results from these in vivo data to patients. I suggest at least to demonstrate synergism of the 4 drug combination in a broader range of doses in order to better prove that low doses are really sufficient to produce a significant anti-tumor effect.

Reviewer #2:

Remarks to the Author:

The authors present an interesting concept of employing combinations of drugs at low exposures to prevent or overcome resistance to TKIs. The combination of TKI, RAF, MEK and PI3K inhibitors shows efficacy in vitro and in vivo and is associated with more pronounced signaling inhibition.

In my opinion, the main challenge with translating this approach to the clinic will be toxicity. Although it is encouraging that this approach does not kill mice or lead to body weight loss, these findings alone are not sufficient to have a real sense of tolerability. Mouse tolerability studies often do not translate to the clinic - they especially fail to demonstrate the skin and gut tox often observed in the clinic with MAPK inhibition; a central tenant of the approach employed by the authors.

In the discussion, the authors note that, "Three-drug combinations given at MTD have been used before in pre-clinical and clinical studies for BRAFV600E mutant tumours, showing clear therapeutic benefits, but such regimen have an associated cost of toxicity." It is probably worth mentioning that those toxicities were not predicted by preclinical studies in mice, highlighting the inability of mice to predict toxicities from MAPK inhibition.

Other comments:

1) For the in vivo studies, the authors use 20% of the reported murine MTDs. Is the PK of any of the individual agents in the combinations affected by the presence of the other agents (via drug-drug interactions). These assessments are required to determine if these doses are truly leading to exposures of each of the individual agents when used in combination as when they are given as single-agents.

2) The authors should determine mechanisms driving resistance in the PC9 ER or OR cell lines.

3) The following statement in the introduction is misleading. "In a meta-analysis of 24 phase 1 clinical studies, patients dosed below agent MTD did almost as well those that received MTD, suggesting that most patients are over-treated with targeted agents¹⁵."

I don't think this meta-analysis warrants such a conclusion for effective targeted therapies, and the authors' conclusions are too simplistic and appear contorted to support the authors approach. For one, most of the 24 trials in that meta-analysis were using medicines that did not have significant activity. Thus, the trial is comparing different dose levels of inactive meds - no wonder there was no benefit to higher doses. I think this statement should be removed.

4) Line 242 - I think it should be supplemental figure 5i.

5) The authors use 0.5mg/kg trametinib for their low-dose studies. However, in our experience 0.3mg/kg matches the exposures observed in the clinic. Indeed, their PK curves show that the exposures achieved in their in vivo studies are higher than the ~30 nM observed with the 2 mg dose in the clinic (Infante, Lancet Oncology, 2012). Thus, the dose of trametinib that is used in the 4D combos results in an exposures that is similar to the MTD of single-agent trametinib in the clinic. Thus, this is not truly low dose trametinib.

Nature Communications Manuscript NCOMMS-19-36009: author's reply to the editor

Reviewer #1 (Remarks to the Author):

The paper by Fernandes Neto et al. illustrates a novel approach to treatment of oncogene-addicted cancer based on the combination of multiple signal transduction inhibitors at low-dose. In particular, this MLD approach showed to be effective in preventing and treating resistance to EGFR tyrosine kinase inhibitors in NSCLC cell lines. Preliminary data of possible activity of similar combinations in other experimental models are also provided. The paper offers novel information that, if validated in a clinical trial, might improve significantly the therapeutic approach in NSCLC patients carrying EGFR mutations. However, there are points that the authors need to clear out in order to improve their manuscript:

1) The authors show in Fig 1C that the combination of different inhibitors has an higher effect than expected based on drug additivity. However, there are specific tools to calculate the additive or synergistic effects of drug combinations that should be used for each of the tested combinations in order to prove that a synergism occurred.

We thank the reviewer for mentioning this point. In the revised manuscript, we performed synergy calculations using the Bliss independence model (according to Russ, D. and Kishony, R. Nat Microbiol, 2018). Figure 1c has been updated to include these results. The main text and figure legend were also updated to accommodate these changes.

2) The authors affirm that target based agents are used in the clinic at their MTD. However, this is not true for all drugs. For example, the MTD of gefitinib, a drug used in this study, was not reached in phase I clinical trials at a dose of 650 mg. In contrast, the recommended dose of gefitinib for EGFR-mutant NSCLC patients is 250 mg. Therefore, this statement must be changed and it should reflect the current standard clinical practice.

We have indeed made a generalization by stating in line 70 that “In these scenarios the drugs are used at maximum tolerated dose (MTD)” and in line 246 that “As such, our data challenge the paradigm that patients should be treated with the MTD of a targeted agent”. We agree with the reviewer that MTD is not the absolute rule for targeted agents. To prevent such generalization, we have now changed the sentences to “In these scenarios the drugs are **usually administered** at maximum tolerated dose (MTD)” and “As such, our data challenge the **common** paradigm that patients should be treated with the MTD of a targeted agent”.

3) The authors need to describe in more details how the PC9 resistant cells were generated. Previous studies have demonstrated that by using different strategies (increasing doses of TKIs, fixed dose of TKI at the IC50 etc...), different mechanisms of resistance are developed by NSCLC cell lines. In addition, some resistant cell lines carry more than one resistance

mechanism. Therefore, a more detailed description of the resistant cell lines might add insights to the interpretation of the results.

We thank the reviewer for mentioning this point. We have now added a more detailed description of how we generated the resistant cell lines in the methods section. In addition, we performed Exome Sequencing of PC9-GR (gefitinib resistant) and PC9-OR (osimertinib resistant) to determine if any *de novo* mutations occurred (Supplemental Table 2). We found the well-known T790M mutation in the GR cells and a number of mutations in the OR cells, none of which has been previously associated with resistance to osimertinib.

4) The data presented in figure 3 are not sufficient to affirm that the combination is not toxic for non-transformed cells. ERK inhibitors have been found to be quite toxic in clinical trials. The concentration of inhibitors in the combination is low. In addition, combinations have been shown to be less toxic as compared with single agents in some cases. However, only a phase I clinical trial will be able to assess the toxicity of this combination that cannot be predicted by an in vitro test. This should be clearly stated by the authors.

We completely agree with the reviewer's comment. We have changed the main text to accommodate this concern. We re-wrote the sentence in line 171 "~~To address test this, we tested~~ the effect of the MLD strategy on **"normal" (non-tumorigenic) cell lines we used in** primary human BJ (fibroblasts) and RPE1 (retinal pigment epithelium) cells." and we also changed in the discussion the paragraph that starts in line 289 "The lack of significant toxicity of the MLD therapy **in mice** may be explained (...) and bone marrow. **However, mice and human are fundamentally different and only a phase I clinical trial will be able to fully assess the toxicity of this strategy in humans.**".

5) The authors show that the low dose combination targeting the EGFR/MAPK signaling cascade is active in PDX models that carry different mechanisms of resistance and, therefore, they conclude that it might be active in all resistant tumors. However, a fraction of EGFR mutant NSCLC have been shown to become resistant to EGFR TKIs by complex mechanisms leading to transformation in SCLC or epithelial-to-mesenchymal transition (EMT). Unless the authors have the possibility to demonstrate that the 4 drugs combination is active in cells carrying such resistance mechanisms, the limits of their approach should be clearly disclosed in the discussion of the results.

We agree with the reviewer's comment. We have studied the histopathology of PDX models used in more detail and found that PDX4 has undergone SCLC transformation. The data to support this notion are shown in the new supplementary figure 5. We have changed the main text to discuss this. We re-wrote the paragraph starting in line 256 "Importantly, we show that tumours (...) KRAS mutation, MET amplification **and even a PDX showing clear hallmarks of SCLC transformation (Supplementary figure 5)**, highlighting that MLD therapy could apply to a diverse range of EGFR TKI resistant tumours. **However, not all possible resistance mechanisms have been tested and it is possible that some might not respond to MLD therapy.**"

6) The results of the in vivo data are in line with the in vitro data. However, I am not sure that these findings really support the hypothesis that multiple low doses are sufficient for tumor growth inhibition. The authors used doses of drugs corresponding to 20-30% of the

MTD for the in vivo studies, although they did not define the specific dose for each drug. If they used 30% of the gefitinib MTD of 10 mg/kg, this corresponds to 3 mg/kg. It means 210 mg for a patient with a weight of 70 kg, which is almost the fixed dose used in the clinic (250 mg). For trametinib, 20% of 0.5 mg/kg is 0.1 mg/kg, i.e 7 mg in a 70 kg patient while the maximum fixed dose approved for treatment of patients is 2 mg. It is evident that the pharmacokinetic and pharmacodynamics of these drugs in mice must be quite different as compared with human. On the other hand, this observation raises several doubts on the possibility to extrapolate results from these in vivo data to patients. I suggest at least to demonstrate synergism of the 4 drug combination in a broader range of doses in order to better prove that low doses are really sufficient to produce a significant anti-tumor effect.

We thank the reviewer for mentioning this point. We have updated the text to make the drug doses used in the in vivo studies more clear: in line 191 “For the in vivo studies **we defined LD as 20% (for gefitinib and trametinib) and 30% (for LY3009120 and SCH772984 - due to the shorter half-lives)** of the published maximum tolerated dose (MTD) in mice for each of the individual drugs^{18,19,29,30}.” Additionally, it was pointed out by the reviewer that some of the drugs, when converted based on mg/kg, were close to the normal concentration that are used in the clinic (hence suggesting that the used concentrations might not really be low doses). As the reviewer has also mentioned, to extrapolate the PK-PD behavior of tyrosine kinase inhibitors from mice to humans is delicate. Interspecies allometric scaling for dose conversion from animal to human studies is one of the most controversial areas in clinical pharmacology. Extrapolation of dose from animals to humans based only in mg/kg can be misleading, since it should be taken into consideration also the body surface area, the pharmacokinetics and the physiological time. To convert the animal dose in mg/kg to human equivalent doses (HED) in mg/kg, it is recommended to divide by 12.3 (see Nair AB, Jacob S. A simple practice guide for dose conversion between animals and human. J Basic Clin Pharm. 2016;7(2):27–31. doi:10.4103/0976-0105.177703). If we estimate the HED based on the low-doses used in our in vivo experiments for Gefitinib and Trametinib (where dosing in humans is known) using this approach:

- Gefitinib 10mg/kg in mice corresponds to 0.810 mg/kg in human, which in a 70kg patient will correspond to 57mg, which is approximately ¼ of the dose used in patients (250mg qd).
- Trametinib 0.5mg/kg in mice corresponds 0,04 mg/kg in human, which in a 70kg patient will correspond to 2,8mg, which is indeed a bit higher than the dose used in patients (2mg qd).

To address this issue we performed an in vivo experiment using lower concentrations of Gefitinib (1mg/kg instead of 10mg/kg) and Trametinib (0.1mg/kg instead of 0.5mg/kg) – Supplemental Figure 4d. **These drug concentrations correspond to 2,5% of the human daily dose for gefitinib and 28% of the daily human dose of trametinib, following the calculation method of Nair mentioned above.** These data indicate that with these reduced concentrations of Gefitinib and Trametinib we still have a significant anti-tumor effect in vivo. We agree that it would be of interest to maybe further test the 4 drug combination in a broader range of doses as the reviewer pointed out. However, due to the difficulty in translating drug doses from mice to human we think this would not bring much more than the already available data. Instead, we believe that the manuscript provides a sound rationale to perform a clinical trial and therefore it is of interest to the patients to move without major delay to a well-designed Phase 1/2 clinical study using the multi low dose

approach. We have included a summary of this mouse-to-human dosing calculation discussion into the main text of the manuscript.

Reviewer #2 (Remarks to the Author):

The authors present an interesting concept of employing combinations of drugs at low exposures to prevent or overcome resistance to TKIs. The combination of TKI, RAF, MEK and PI3K inhibitors shows efficacy in vitro and in vivo and is associated with more pronounced signaling inhibition.

In my opinion, the main challenge with translating this approach to the clinic will be toxicity. Although it is encouraging that this approach does not kill mice or lead to body weight loss, these findings alone are not sufficient to have a real sense of tolerability. Mouse tolerability studies often do not translate to the clinic - they especially fail to demonstrate the skin and gut tox often observed in the clinic with MAPK inhibition; a central tenant of the approach employed by the authors.

In the discussion, the authors note that, "Three-drug combinations given at MTD have been used before in pre-clinical and clinical studies for BRAFV600E mutant tumours, showing clear therapeutic benefits, but such regimen have an associated cost of toxicity." It is probably worth mentioning that those toxicities were not predicted by preclinical studies in mice, highlighting the inability of mice to predict toxicities from MAPK inhibition.

We thank the reviewer for the comments. As mentioned by the reviewer one of the major problems in cancer research is the difficulty in translating in vitro/vivo studies to human studies, particularly regarding toxicity issues. We acknowledge that often toxicities can't be fully predicted by preclinical studies in mice. We tried, in the best of our abilities, to predict such possibility by analyzing the weight of the mice over time and by doing histological analysis of several organs (brain, heart, liver, spleen, bone marrow and GI tract) after the mice were sacrificed. We did not detect any signs of toxicity. However, mice and human are fundamentally different and only a phase I clinical trial will be able to fully assess the toxicity of this strategy in humans. We have indicated this notion in the revised text to emphasize this point.

Other comments:

1) For the in vivo studies, the authors use 20% of the reported murine MTDs. Is the PK of any of the individual agents in the combinations affected by the presence of the other agents (via drug-drug interactions). These assessments are required to determine if these doses are truly leading to exposures of each of the individual agents when used in combination as when they are given as single-agents.

We thank the reviewer for mentioning this point. To address this, we have performed an additional PK study where we treated mice with the individual drugs or with the 4D combination. We then analyzed if the half-lives were different in the 2 conditions (Supplemental Figure 6e). Overall, there is not any apparent drug-drug interaction, except for LY3009120, for which the half-life increases when given in the 4D Combo. The main text was adjusted to include this new data.

2) The authors should determine mechanisms driving resistance in the PC9 ER or OR cell lines.

In the revised manuscript, we have added a more detailed description of how we generated the resistant cell lines in the methods section. In addition, we performed Exome Sequencing of PC9-GR (gefitinib resistant) and PC9-OR (osimertinib resistant) to determine if any *de novo* mutations occurred (Supplemental Table 2). We found the well-known T790M mutation in the GR cells and a number of mutations in the OR cells, none of which has been previously associated with resistance to osimertinib.

3) The following statement in the introduction is misleading. “In a meta-analysis of 24 phase 1 clinical studies, patients dosed below agent MTD did almost as well those that received MTD, suggesting that most patients are over-treated with targeted agents¹⁵.” I don’t think this meta-analysis warrants such a conclusion for effective targeted therapies, and the authors’ conclusions are too simplistic and appear contorted to support the authors approach. For one, most of the 24 trials in that meta-analysis were using medicines that did not have significant activity. Thus, the trial is comparing different dose levels of inactive meds – no wonder there was no benefit to higher doses. I think this statement should be removed.

We agree with the reviewer’s comment and we have therefore removed this sentence from the manuscript.

4) Line 242 – I think it should be supplemental figure 5i.

There is indeed a mistake in the figure numbering here. It should be Figure 4h. This is now corrected in the main text.

5) The authors use 0.5mg/kg trametinib for their low-dose studies. However, in our experience 0.3mg/kg matches the exposures observed in the clinic. Indeed, their PK curves show that the exposures achieved in their in vivo studies are higher than the ~30 nM observed with the 2 mg dose in the clinic (Infante, Lancet Oncology, 2012). Thus, the dose of trametinib that is used in the 4D combos results in an exposures that is similar to the MTD of single-agent trametinib in the clinic. Thus, this is not truly low dose trametinib.

We thank the reviewer for mentioning this point. At the recommended dose in humans (2mg qd), the AUC was 370 (ng h¹ mL¹) at day 15, based on the PK results reported in the phase I study by Infante, Lancet Oncol 2012. In the trametinib NDA submitted to FDA, an experiment using athymic nude mice reported that the AUC corresponding to 0.1, 0.3 and 1 mg/kg/day at day 14 were 347, 1431 and 6785. If we take into account the HED rule (see Nair AB, Jacob S. A simple practice guide for dose conversion between animals and human. J Basic Clin Pharm. 2016;7(2):27–31. doi:10.4103/0976-0105.177703) this would translate the AUC into approximately 28, 116 and 552. If we do a linear regression analysis, the dose in mice that would correspond to 370 (ng h¹ mL¹) in humans would be ~0.7 mg/kg, which is slightly above the 0.5mg/kg used in our study. We therefore agree with the reviewer’s comment that the trametinib dose used in the majority of the in vivo experiments in the manuscript is similar to the MTD of single-agent trametinib. To address this issue we

performed an in vivo experiment using lower concentrations of Gefitinib (1mg/kg instead of 10mg/kg) and Trametinib (0.1mg/kg instead of 0.5mg/kg),– Supplemental Figure 4d. **These drug concentrations correspond to 2,5% of the human daily dose for gefitinib and 28% of the daily human dose of trametinib, following the calculation method of Nair mentioned above.** These data indicate that even with reduced concentrations of Gefitinib and Trametinib we still have a significant anti-tumor effect, although to a lesser extent than with the higher doses. We have included a summary of this mouse-to-human dosing calculation discussion into the main text of the manuscript.

Reviewers' Comments:

Reviewer #1:

Remarks to the Author:

The authors have addressed the points that I raised and the manuscript is significantly improved.

Reviewer #3:

Remarks to the Author:

Discussion,

However, mice and human are fundamentally different with respect to drug toxicity and only a phase I clinical trial will be able to fully assess the toxicity of this strategy in humans.

Reviewer #2 was specifically calling out the lack of correlation between mouse and human skin toxicity with MAP kinase pathway targeted therapeutics. This more general statement doesn't acknowledge that point. The toxicity of MEK and ERK inhibitors, and even MEK/ERK combination, have long been underestimated in mice, leading to far less impressive efficacy in humans when these drugs are given at maximum tolerated doses.

The authors have been responsive to the other critiques.

Response to reviewers manuscript NCOMMS-19-36009A

Reviewer #1 (Remarks to the Author):

The authors have addressed the points that I raised and the manuscript is significantly improved.

Reviewer #3 (Remarks to the Author):

Discussion,

However, mice and human are fundamentally different with respect to drug toxicity and only a phase I clinical trial will be able to fully assess the toxicity of this strategy in humans.

Reviewer #2 was specifically calling out the lack of correlation between mouse and human skin toxicity with MAP kinase pathway targeted therapeutics. This more general statement doesn't acknowledge that point. The toxicity of MEK and ERK inhibitors, and even MEK/ERK combination, have long been underestimated in mice, leading to far less impressive efficacy in humans when these drugs are given at maximum tolerated doses.

We agree with the comments made by the reviewer. We have adjusted the discussion section to reflect specifically the notion that skin toxicity cannot be measured in mouse experiments.

The revised discussion now reads: *However, mice and human are fundamentally different with respect to drug toxicity. Especially skin toxicity following MAPK inhibition therapy is often underestimated in mice. Therefore, only a phase I clinical trial will be able to fully assess the toxicity of this strategy in humans.*

The authors have been responsive to the other critiques.